# Rare CIDEC coding variants enriched in age-related macular degeneration patients with small low-luminance deficit cause lipid droplet and fat storage defects

**Sehyun Kim**[1¤a], **Amy Stockwell**[2], **Han Qin**[1], **Simon S. Gao**[3], **Meredith Sagolla**[4], **Ivaylo Stoilov**[5], **Arthur Wuster**[2¤b], **Phillip Lai**[6], **Brian L. Yaspan**[2☯‡], **Marion Jeanne**[1☯‡*]

1 Department of Neuroscience, Genentech Inc., South San Francisco, California, United States of America,
2 Department of Human Genetics, Genentech Inc., South San Francisco, California, United States of America, 3 Department of Clinical Imaging, Genentech Inc., South San Francisco, California, United States of America, 4 Department of Research Pathology, Genentech Inc., South San Francisco, California, United States of America, 5 Medical Affairs Ophthalmology, Genentech Inc., South San Francisco, California, United States of America, 6 Early Clinical Development, Genentech Inc., South San Francisco, California, United States of America

☯ These authors contributed equally to this work.
¤a Current address: ABL Bio Inc., Seongnam, Republic of Korea
¤b Current address: Department of Translational Genomics, BioMarin Pharmaceutical, San Rafael, California, United States of America
‡ BLY and MJ are joint senior authors to this work.
* jeanne.marion@gene.com

**Data Availability Statement:** All reagents used in this study are commercially available and supplier names/catalog numbers are provided in the

## Abstract

### Background

The basis of Age-related macular degeneration (AMD) genetic risk has been well documented; however, few studies have looked at genetic biomarkers of disease progression or treatment response within advanced AMD patients. Here we report the first genome-wide analysis of genetic determinants of low-luminance vision deficit (LLD), which is seen as predictive of visual acuity loss and anti-VEGF treatment response in neovascular AMD patients.

### Methods

AMD patients were separated into small- and large-LLD groups for comparison and whole genome sequencing was performed. Genetic determinants of LLD were assessed by common and rare variant genetic analysis. Follow-up functional analysis of rare coding variants identified by the burden test was then performed *in vitro*.

### Results

We identified four coding variants in the *CIDEC* gene. These rare variants were only present in patients with a small LLD, which has been previously shown to indicate better prognosis and better anti-VEGF treatment response. Our *in vitro* functional characterization of these *CIDEC* alleles revealed that all decrease the binding affinity between CIDEC and the lipid

Materials and Methods section of the manuscript. Human subjects were part of the HARBOR clinical trial, ClinicalTrials.gov identifier: NCT00891735, and the study population has been previously described for low-luminance deficit (LLD): Frenkel, R.E., Shapiro, H., and Stoilov, I. (2016). Predicting vision gains with anti-VEGF therapy in neovascular age-related macular degeneration patients by using low-luminance vision. The British journal of ophthalmology 100, 1052-1057 http://doi.org/10.1136/bjophthalmol-2015-307575 Individual genetic data and other privacy-sensitive individual information are not publicly available because they contain information that could compromise research participant privacy. All publicly available code and software has been identified in the methods section of the manuscript. We are unable to share genome-wide individual level data, even de-identified, due to restrictions on the patient consents, however, all the summary statistics for the genetics analysis can be provided upon request to the corresponding author (Dr Marion Jeanne: jeanne.marion@gene.com) and/or the lead Human Geneticist (Dr Brian Yaspan: yaspan.brian@gene.com). Data is available for qualified researcher employed or legitimately affiliated with an academic, non-profit or government institution who have a track record in the field. We would ask the researcher to sign a data access agreement that needs to be signed by applicants and legal representatives of their institution, as well as legal representatives of Genentech, Inc. A brief research proposal will be needed to ensure that 'Applications for access to Data must be Specific, Measurable, Attainable, Resourced and Timely.' The following previously published datasets were used: 1. Human Retina and RPE/Choroid bulk RNA sequencing, data from: Orozco, L.D., Chen, H.H., Cox, C., Katschke, K.J., Jr., Arceo, R., Espiritu, C., Caplazi, P., Nghiem, S.S., Chen, Y.J., Modrusan, Z., et al. (2020). Integration of eQTL and a Single-Cell Atlas in the Human Eye Identifies Causal Genes for Age-Related Macular Degeneration. Cell Rep 30, 1246-1259 e1246. https://doi.org/10.1016/j.celrep.2019.12.082 2. Human eye single cell RNA sequencing, data from: Gautam, P., Hamashima, K., Chen, Y., Zeng, Y., Makovoz, B., Parikh, B.H., Lee, H.Y., Lau, K.A., Su, X., Wong, R.C.B., et al. (2021). Multi-species single-cell transcriptomic analysis of ocular compartment regulons. Nat Commun 12, 5675. https://doi.org/10.1038/s41467-021-25968-8.

**Funding:** No external funding was obtained for this study. At the time of the study, all authors were full time employees of Genentech/Roche. The study was funded by general Genentech/Roche funding. The funders had no role in study design, data

droplet fusion effectors PLIN1, RAB8A and AS160. The rare *CIDEC* alleles all cause a hypomorphic defect in lipid droplet fusion and enlargement, resulting in a decreased fat storage capability in adipocytes.

## Conclusions

As we did not detect CIDEC expression in the ocular tissue affected by AMD, our results suggest that the *CIDEC* variants do not play a direct role in the eye and influence low-luminance vision deficit via an indirect and systemic effect related to fat storage capacity.

## Introduction

Age-related macular degeneration (AMD) accounts for nearly 10% of blindness worldwide, and is the leading cause of blindness in developed countries [1]. AMD is a progressive retinal disease characterized by the accumulation of extracellular deposits called drusen, underneath the retina in the early stages of the disease, followed by either atrophy of the macula in the advanced dry form of AMD called Geographic Atrophy (GA), and/or growth of pathogenic blood vessels into the retina in the wet form of AMD called neovascular AMD. Both GA and neovascular AMD are clinical end-stages forms of AMD and lead to progressive and severe vision loss. There is currently no approved treatment for GA and despite anti-Vascular Endothelial Growth Factor (VEGF) intraocular injections having revolutionized the treatment of neovascular AMD, they are not curative and patient response is heterogeneous [2].

Although the pathophysiology of AMD is still not completely understood, there is a well-established genetic component to disease risk. Concordance rates between mono-zygotic twins are significantly higher than di-zygotic twins [3–5]. Both population-based and familial studies have found evidence of sibling correlations, and estimate that genetic factors can account for between 50% and 70% of the total variability in disease risk [6, 7]. Furthermore, it is estimated that genetic risk factors account for up to 71% of variation in the severity of disease [8]. Genome-wide association studies (GWAS) of AMD disease risk have greatly expanded our knowledge around the disease and especially its biology, with the most recent study involving over 16,000 AMD patients and 17,000 controls finding 52 independently associated variants [9]. Major risk loci identified include complement genes (e.g. *CFH*, *CFI*, *C3*, *C9*) and the *ARMS2/HTRA1* locus. However, there are several other pathways identified including genes involved in lipid metabolism (e.g. *LIPC*, *CETP*) and extracellular matrix remodeling (e.g. *TIMP3*, *MMP9*).

It is known that subjects with AMD have difficulty seeing in dimly lit environments [10]. As such, the reduction in visual acuity under suboptimal illumination known as low-luminance deficit (LLD) has been evaluated in AMD patients and is seen to be predictive of both the development of GA with subsequent visual acuity loss and response to anti-VEGF treatment in neovascular AMD patients [11, 12].

Here we report the first genome-wide investigation into genetic determinants for low-luminance dysfunction in neovascular AMD utilizing patient data from the HARBOR (ranibizumab), TENAYA and LUCERNE (faricimab) phase 3 clinical trials [13, 14]. The HARBOR trial was a dosing study which sought to determine the efficacy and safety of 2.0 mg and 0.5 mg doses of ranibizumab (anti-VEGF antibody) in treatment naive patients with choroidal neovascularization (CNV) secondary to AMD [13, 15]. This study enrolled 1098 patients and followed them for one year. All dosing groups demonstrated clinically meaningful visual

collection and analysis, decision to publish, or preparation of the manuscript.

**Competing interests:** I have read the journal's policy and the authors of this manuscript have the following competing interests: at the time of the study, all authors were full time employees of Genentech/Roche with stock and stock options in Roche. This does not alter our adherence to PLOS ONE policies on sharing data and materials.

**Abbreviations:** AMD, Age-related macular degeneration; AR, autosomal recessive; BCVA, best corrected visual acuity; CIDEC, Cell-death-Inducing DNA fragmentation factor (DFF)45-like Effector C; CNV, choroidal neovascularization; EST, Expressed Sequence Tag; FPLD5, Familial Partial Lipodystrophy type 5; GA, Geographic Atrophy; GWAS, Genome-wide association studies; IAMDGC, International AMD Genetics Consortium; LD, lipid droplet; LLD, low-luminance deficit; LLVA, low-luminance visual acuity; MAF, minor allele frequency; MOI, mean optical intensity; OCT, Optical Coherence Tomography; OR, odds ratio; Q, quartile; RGCs, Retinal Ganglion Cells; RPE, Retinal Pigment Epithelium; SNP, single-nucleotide polymorphism; VEGF, Vascular Endothelial Growth Factor; WGS, whole genome sequencing; WT, wild-type.

improvement. TENAYA and LUCERNE were randomised, double-masked, non-inferiority trials designed to investigate the efficacy, durability, and safety of intravitreal faricimab, a bispecific antibody acting through dual inhibition of angiopoietin-2 and VEGF A. The trials enrolled 1329 patients and followed them for two years. Multiple clinical datapoints were collected at baseline, including LLD. We separated the trial patients into two groups for comparison, those with the largest LLD differential (biggest drop in vision under low-luminance, quartile 4 = Q4) and those with the smallest LLD differential before ranibizumab treatment (quartile 1 = Q1). HARBOR was used as a discovery dataset, and any associations of interest were replicated in TENAYA and LUCERNE. We selected phenotypic extremities instead of the whole patient population for two main reasons; (1) the data looking at the effect of baseline LLD on anti-VEGF treatment response showed the largest difference between Q1 and Q4 patients [12] and (2) it has been suggested as a way to increase power in genetic studies [16]. Because the genetic underpinnings of LLD differential has not been fully explored, we entered the study with the goal of identifying genetic factors involved in LLD using common and rare variation assayed via whole genome sequencing (WGS) with functional follow-up of biologically interesting hits. For functional characterization, we then selected from the top hits the *CIDEC* gene as a compelling candidate gene with reported function related to lipid metabolism, a pathway identified in previous AMD genetic analyses [9].

## Results

### A genome wide burden test identifies rare genetic variants in the *CIDEC* gene that are enriched in AMD patients with small low-luminance deficit

For this study, we first subset the HARBOR ranibizumab dosing study population as previously described for baseline low-luminance deficit (LLD) [12]. All patients in the HARBOR trial had neovascular AMD. This subsetting resulted in 275 patients in the Q1 population, and 241 patients in the Q4 population (Fig 1A). Detailed population characteristics are seen in Table 1. We compared LLD quartiles 1 (Q1) and 4 (Q4) for this analysis, with the goal of maximizing the phenotypic difference as seen in the previous anti-VEGF treatment response study [12]. Patients in Q1 (smallest low-luminance deficit) were seen to have better outcome on anti-VEGF therapy, and slower visual acuity loss in GA patients than patients in Q4 (large low-luminance deficit) [11, 12]. In the study population, patients in Q1 were more likely to have lower baseline visual acuity, smaller baseline CNV leakage area, thinner sub-retinal fluid and a thinner choroid, but did not significantly differ by age or sex (Table 1). We coded Q1 as the "cases" and Q4 as the "controls", so subsequently an odds ratio (OR) >1 indicates the minor allele was enriched in Q1 and an OR <1 indicates the minor allele was enriched in Q4.

The lines of genetic investigation are outlined in Fig 1B. We first investigated the loci identified in a recent AMD risk meta-analysis from the International AMD Genetics Consortium (IAMDGC) (S1 Table in S1 File) [8]. Twenty-four single-nucleotide polymorphisms (SNPs) identified in the IAMDGC study were available for analysis. No locus retained statistical significance after multiple testing. Two loci had $P < 0.1$, (1) *ARHGAP21*, rs12357257, (odds ratio (OR) = 0.63, p = 0.004) and (2) *LIPC*, rs2043085, (OR = 1.29, P = 0.10). We also constructed polygenic risk scores (PRS) for 1) advanced AMD risk 2) neovascular AMD risk and 3) geographic atrophy risk from the same IAMDGC consortium analysis. We did not find any of these PRS to be associated with our LLD population. Next, we examined common variation throughout the genome (SNPs with a minor allele frequency (MAF) > 0.01). There were no SNP which met the genome-wide significance level of $p < 5 \times 10^{-8}$ (S1 Fig in S1 File).

We then evaluated rare variation (SNPs with a MAF<0.01) in the form of a burden test in the discovery dataset. We included exonic SNPs predicted to have a moderate (e.g. amino acid

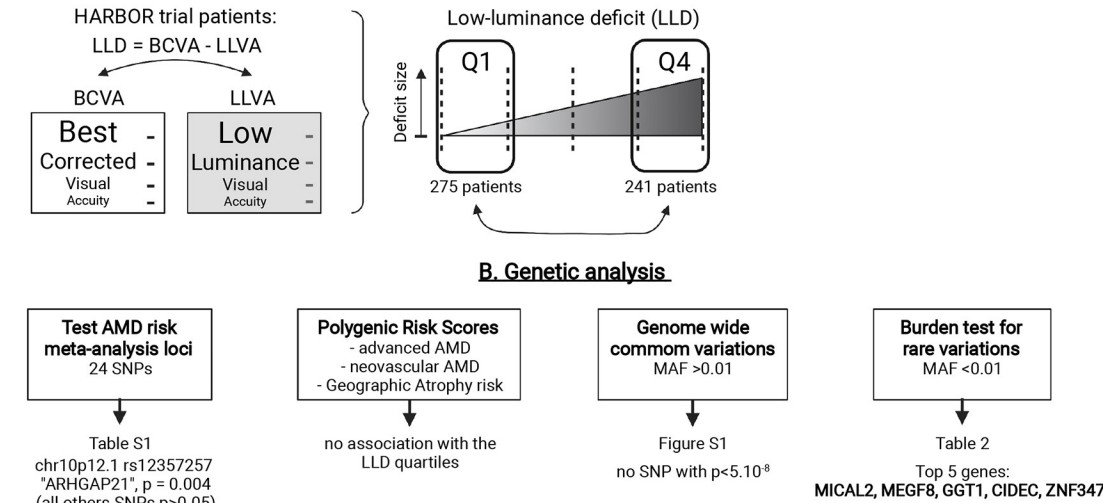

## C. Gene prioritization for functional follow-up

| | Biological function | Mutation associated with disease reported (OMIM)? | Mostly expressed in? | Expressed in the eye? | Causal gene in the locus? | RV associated with Q1? | Biological pathway associated with AMD? | Biomarker opportunity? |
|---|---|---|---|---|---|---|---|---|
| chr10p12.1 rs12357257 "ARHGAP21" | N/A | no | N/A | N/A | unknown | N/A | N/A | no |
| MICAL2 | F-actin disassembly protein | no | Brain, Artery | yes (RGCs) | yes, coding variants | yes | no | unlikely |
| MEGF8 | negative regulator of Hedgehog signaling | yes, AR, Carpenter syndrome (OMIM 614976) | Brain | yes (RGCs) | yes, coding variants | yes | no | unlikely |
| GGT1 | cystein/glutathione homeostasis | yes, AR, Glutathionuria (OMIM 231950) | Kidney, Liver | very low | yes, coding variants | yes | no | yes |
| CIDEC | lipid metabolism | yes, AR, FPLD5 (OMIM 615238) | Adipose tissue | no | yes, coding variants | yes | yes | yes |
| ZNF347 | transcriptional regulation (?) | no | Brain | low | yes, coding variants | no | no | unlikely |

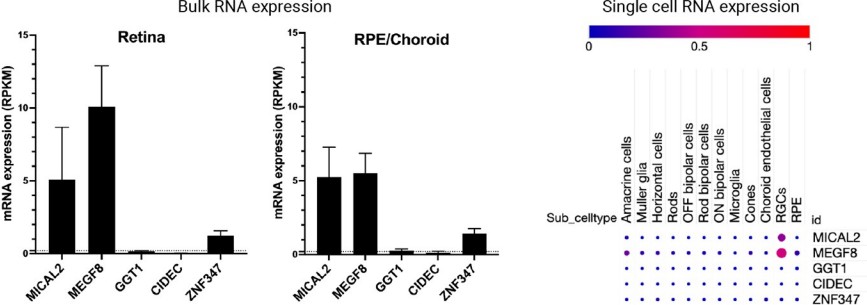

**Fig 1. Overview of the patient stratification, the lines of genetic investigation performed and the strategy used to prioritize genes for functional follow-up.** (A). HARBOR patients were separated at baseline into two groups based on the size of their Low-Luminance Deficit (LLD): patients in quartile 1 (Q1) had the smallest drop in vision under low-luminance and patients in quartile 4 (Q4) had the biggest deficit. (B). Lines of genetic investigation and top-line results. (C) For functional analysis follow-up, top genetic hits were prioritized based on different criteria such as being the causal gene at the locus (presence of coding variants), the rare variants (RV) identified being enriched in Q1 patients, the gene playing a role in a biological pathway associated with AMD pathophysiology, and providing a potential biomarker opportunity. For the top hits, gene expression in human retina or RPE/choroid (bulk RNA sequencing, data from Orozco et al. [17]) and in different human ocular cell types (single cell RNA sequencing, data from Gautam et al. [18]) were also analyzed. AR: autosomal recessive; FPLD5: Familial Partial Lipodystrophy type 5. RGCs: Retinal Ganglion Cells. RPE: Retinal Pigment Epithelium.

**Table 1. Quartile Q1 and quartile Q4 AMD HARBOR patient demographic comparison.**

| | Q1 | Q4 | p value | Missing (N%) | |
| --- | --- | --- | --- | --- | --- |
| | | | | Q1 | Q4 |
| N | 275 | 241 | | | |
| Age | 78.61 (9.07) | 78.85 (7.94) | 0.75 | 0 (0%) | 0 (0%) |
| Female, N (%) | 117 (43%) | 105 (44%) | 0.81 | 0 (0%) | 0 (0%) |
| Baseline visual acuity | 48.0 (14.4) | 57.6 (9.2) | 5.12E-15 | 0 (0%) | 0 (0%) |
| Baseline CNV leakage area | 2.98 (1.83) | 4.35 (2.25) | 9.53E-12 | 0 (0%) | 0 (0%) |
| Baseline sub-retinal fluid thickness | 98 (95) | 172 (125) | 1.09E-11 | 0 (0%) | 0 (0%) |
| Baseline choroidal thickness | 174 (58) | 200 (77) | 0.0023 | 116 (42.2%) | 111 (46%) |

changing) to high (e.g. stop codon gain or loss) impact on the final protein sequence. No loci identified in the recent GWAS meta-analysis were significantly associated in our burden test (all p>0.05). No gene burden test passed a Bonferroni multiple testing cutoff for the number of genes in the genome tested. The top five hits from the discovery analysis are presented in Table 2.

As a replication population was not yet available, we prioritized the top genetic hits identified by our common and rare variants analysis using different criteria (Fig 1C) for functional analysis follow-up. We decided to select *CIDEC* for a thorough wet lab analysis as it was the probable causal gene at the identified locus (presence of coding variants), the rare variants identified were enriched in Q1 patients (i.e. associated with better outcome). Furthermore, CIDEC is involved in a biological pathway already associated with AMD (i.e. lipid metabolism) and since CIDEC expression is broad in the human body (adipose tissue), it provides a potential biomarker opportunity [19], which is usually not the case when the gene expression is restricted to the neuroretina.

The *CIDEC* gene encodes the CIDEC protein (NP_001365420.1; OMIM: 612120), a member of the Cell-death-Inducing DNA fragmentation factor (DFF)45-like Effector (CIDE) family. The *CIDEC* rare alleles found in the discovery analysis were found in 6% of Q1 patients and spread over multiple exons. In the Q4 patients, rare alleles were found in 2% of individuals in the form of one SNP (rs140125102). This SNP was located in one exon seen only in RefSeq transcript NM_001199551 (Fig 2A). We sought to quantify the percentage of transcripts expressed that are NM_001199551 in the GTEx database for adipose tissue and blood [20]. In both sample types, percent expression of NM_001199551 was 0.5% of all *CIDEC* transcripts (S2 Fig in S1 File–adipose pictured, blood similar). In conclusion, if restricting the analysis in *CIDEC* to exons contained in transcripts that are more widely expressed we found that *CIDEC* rare variation was exclusive to the Q1 AMD patients (N = 12).

**Table 2. Results from rare variant burden test comparing quartile Q1 and quartile Q4 AMD HARBOR patients.**
Frequencies in the columns denote the fraction of individuals in that cohort who possess any rare, functionally predicted variant identified in the study. OR = odds ratio.

| Gene | Q1 Freq | Q4 Freq | # SNPs | OR | p value |
| --- | --- | --- | --- | --- | --- |
| *MICAL2* | 0.14 | 0.06 | 28 | 3.38 | 7.70E-04 |
| *MEGF8* | 0.12 | 0.05 | 27 | 3.75 | 9.77E-04 |
| *GGT1* | 0.16 | 0.08 | 24 | 2.88 | 1.06E-03 |
| *CIDEC* | 0.06 | 0.02 | 5 | 7.14 | 1.08E-03 |
| *ZNF347* | 0.02 | 0.10 | 15 | 0.20 | 1.68E-03 |

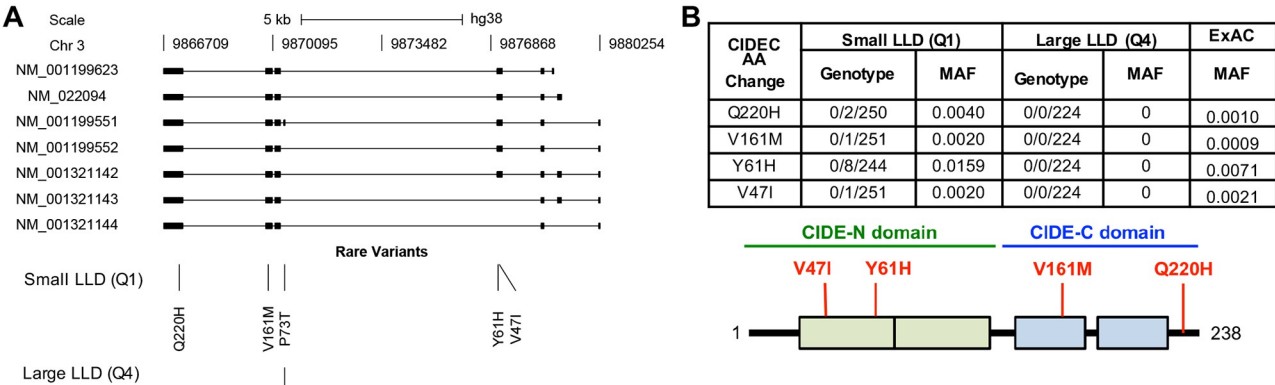

**Fig 2. Genetic analysis of low-luminance deficit quartile Q1 and quartile Q4 AMD patients.** (A) Genetic diagram of *CIDEC* and location of rare variants in Q1 and Q4 AMD patients. SNPs are indicated by amino acid change and position. (B) Table of genotype and minor allele frequencies for variants selected for further analysis and map of CIDEC protein with CIDE-N and CIDE-C domains with these SNPs annotated by position and amino acid change.

The four SNPs identified in Q1 patients were rs150971509 c.139G>A [p.Val47Ile], rs79419480 c.181T>C [p.Tyr61His], rs145323356 c.481G>A [p.Val161Met] and rs52790883 c.660G>T [p.Gln220His] (subsequently referred to as V47I, Y61H, V161M and Q220H respectively) (Fig 2B). Three of these SNPs (V47I, V161M and Q220H) are rare (minor allele frequency (MAF) <0.005) in all super-populations seen in gnomAD. Y61H is rare in the European and East Asian super populations, but common (MAF>0.01; range 0.01–0.07) in others (South Asian, Ashkenazi Jewish, African/African American and Latinx/Admixed American), with the frequency in the South Asian population being the greatest (MAF = 0.07) [21]. We used the software PolyPhen-2 (Polymorphism Phenotyping v2) to perform *in silico* prediction of the possible impact of these four amino acid substitutions on CIDEC stability or function [22]. The V47I substitution was predicted as probably damaging, the V161M and Q220H substitutions were predicted as possibly damaging and only the Y61H substitution was predicted to be benign. Since no structural data was available for the full CIDEC protein, these predictions were based solely on evolutionary comparisons. Thus, we decided to include the four rare variants identified in our Q1 AMD CIDEC discovery population patients in our experimental follow-up.

While functional follow-up was underway, we identified a suitable replication study population in the TENAYA and LUCERNE neovascular AMD studies. We subset the TENAYA and LUCERNE study populations in the same manner as described for HARBOR. This resulted in 171 patients in the Q1 population and 170 patients in the Q4 population (S2 Table in S1 File). No gene replicated at p>0.05. The top p value for the replication was for *CIDEC* (with the aforementioned SNP rs140125102 removed, seen in 1 Q1 patient and 3 Q4 patients), with the same direction of effect as in the discovery analysis (OR = 3.61; p = 0.14). We observed *CIDEC* variants in 4% of the Q1 population and 1% of the Q4 population. When combined with the discovery analysis, *CIDEC* was the top hit (OR = 5.71; p = 4.1x10$^{-4}$). When removing SNP rs140125102 from the discovery population, the effect size notably increased (OR = 15.92; p = 8.0x10$^{-4}$). In this combined analysis of 424 Q1 patients and 397 Q4 patients, *CIDEC* rare variants appeared in 4% of the Q1 population and 0.5% of the Q4 population. Thus, both our ad-hoc prioritization and replication analysis pointed to *CIDEC* as the best candidate gene for functional follow-up (Table 3).

**Table 3. Results from rare variant burden test comparing quartile Q1 and quartile Q4 in a combined analysis (meta) of AMD patients from the HARBOR (discovery), TENAYA and LUCERNE (replication) trials.** OR = odds ratio.

| Gene | HARBOR | | TENAYA and LUCERNE | | Meta analysis | |
| --- | --- | --- | --- | --- | --- | --- |
| | Q1 Freq | Q4 Freq | Q1 Freq | Q4 Freq | OR | p value |
| *CIDEC* | 0.06 | 0.02 | 0.04 | 0.01 | 5.71 | 4.10E-04 |
| *GGT1* | 0.16 | 0.08 | 0.01 | 0.00 | 2.88 | 1.05E-03 |
| *MICAL2* | 0.14 | 0.06 | 0.09 | 0.07 | 2.35 | 1.54E-03 |
| *ZNF347* | 0.02 | 0.10 | 0.03 | 0.03 | 0.42 | 0.27 |
| *MEGF8* | 0.12 | 0.05 | 0.10 | 0.09 | 1.92 | 0.32 |

## Q1 AMD CIDEC rare alleles cause a defect in lipid droplet fusion and enlargement

*CIDEC* is a homolog of the murine *Fsp27* (Fat-specific protein 27kDa) gene [23]. *Fsp27* was originally identified as a gene up-regulated during murine pre-adipocytes differentiation *in vitro* [24, 25]. FSP27 was then shown to localize to lipid droplets (LDs) in adipocytes, where it promotes triglyceride storage by inhibiting LD fragmentation and lipolysis [26]. *In vivo*, FSP27 is mainly expressed in the white adipocytes where it contributes to optimal energy storage by allowing the formation of their characteristic large unilocular LD [27]. *Fsp27* deficient mice have white adipocytes with small multilocular LDs and increased mitochondrial size and activity, resulting in smaller white fat pads and increased metabolic rate [27, 28]. A *CIDEC* homozygous nonsense mutation was identified in a patient with partial lipodystrophy and insulin resistant diabetes (OMIM: 615238) [29]. This p.Glu186* (E186X, c.556G → T) mutation results in truncation of the CIDEC protein and the patient presented with multilocular small LDs and focal increased mitochondria density in adipocytes. Notably, *Fsp27* deficient mice have a healthy metabolic profile but when challenged by substantial energetic stress, they acquire features found in the CIDEC E186X patient, such as insulin resistance and hepatic steatosis [30]. However, no eye phenotype has been reported in the CIDEC E186X patient nor the *Fsp27* deficient mice. Therefore, we first investigated the potential functional consequences of the four rare, protein altering CIDEC alleles found in Q1 AMD patients in adipocytes, a cell type in which CIDEC's function has been well established.

First, we transiently expressed different versions of CIDEC tagged with GFP into 3T3-L1 pre-adipocytes. We transfected each of the four Q1 AMD rare variants (V47I, Y61H, V161M and Q220H) and as controls, we transfected cells with CIDEC wild-type (WT) or with the CIDEC E168X mutation. Subsequently, the proteins encoded by the Q1AMD rare *CIDEC* alleles will be referred to as "AMD CIDEC variants". The cells were then treated for two days with oleic acid to induce LD formation. As expected, the mutant CIDEC E168X was diffused in the cytoplasm and failed to accumulate around the LDs (data not shown, and [29]). In contrast, the four AMD CIDEC variants mostly localized to LDs in transfected adipocytes, and similarly to CIDEC WT, accumulated at the LD-LD contact sites (Fig 3A).

Next, we assessed the size of the LDs in the transfected adipocytes. We found that cells transfected with CIDEC WT had LDs with an average diameter of 2 to 3 μm. However, cells expressing CIDEC E168X had a severe LD enlargement defect, resulting in accumulation of clustered LDs with diameters smaller than 1 μm. Cells expressing each of the AMD CIDEC variants had an intermediate phenotype with a majority of LDs being smaller than 2 μm (Fig 3B and 3C). Interestingly, unlike in the E168X mutation case or *Fsp27* deficiency, we found that the presence of the AMD CIDEC variants did not increase the density of mitochondria in the transfected cells (S3A Fig in S1 File) and they did not alter mitochondria

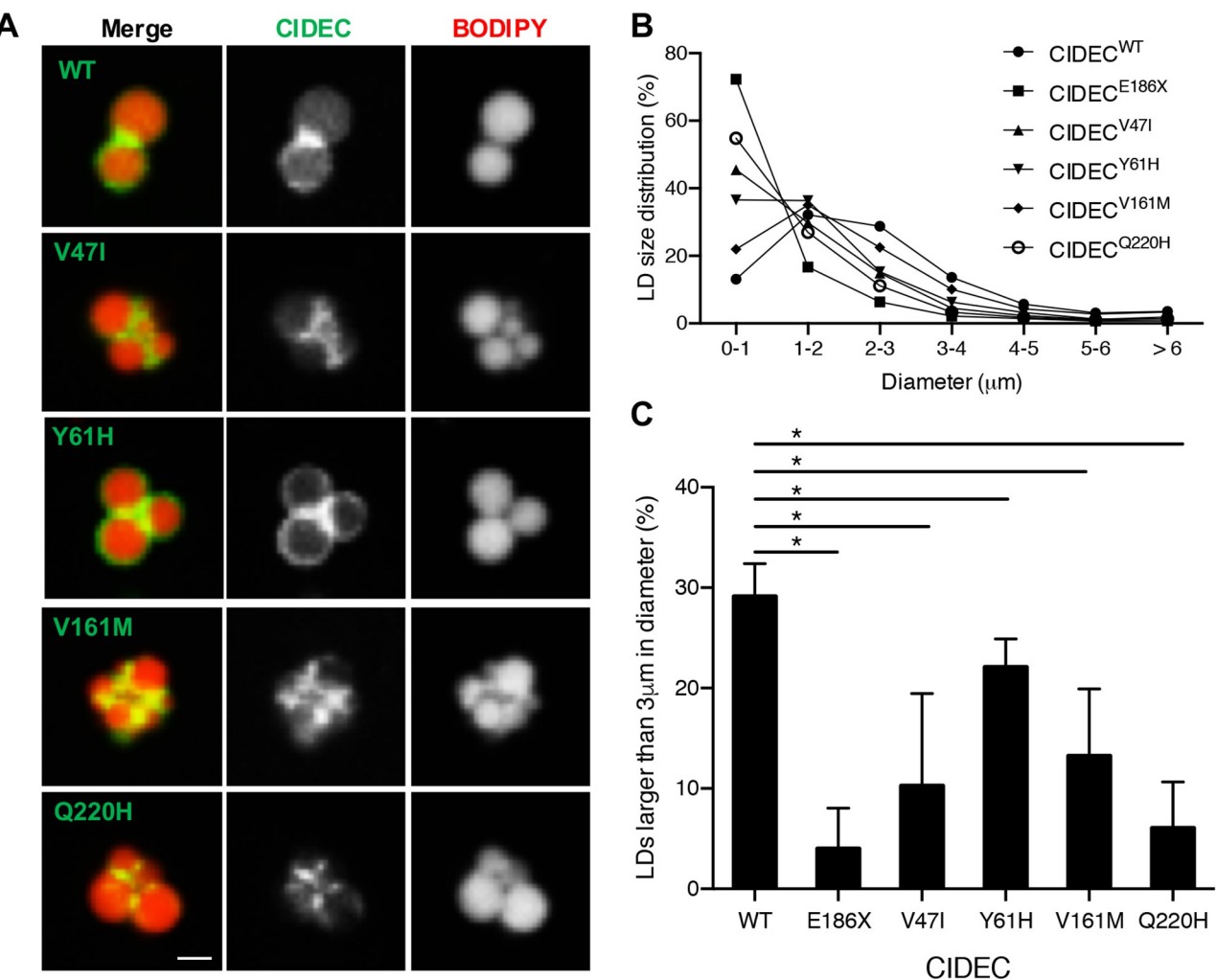

**Fig 3. AMD CIDEC variants localize to lipid droplets (LDs) but cause a defect in LD enlargement.** (A) Representative images of GFP-tagged CIDEC wild-type (WT) or rare variants localized to LDs labeled in red by BODIPY 558/568. Scale bar: 2 μm. (B) Size distribution of LDs in pre-adipocytes expressing CIDEC WT or each of the rare variants (diameters in μm). (C) Percentage of LDs with a diameter larger than 3 μm. N = 3 (mean ± SD, Student's t test, *p<0.05).

activity as measured with a Seahorse bioanalyzer (S3B Fig in S1 File). In conclusion, the AMD CIDEC variants do not impair proper CIDEC localization to LDs and do not increase mitochondrial density, but they are hypomorphic variants reducing the LD enlargement capacity in adipocytes.

Next, we transiently co-expressed GFP-tagged version of CIDEC WT or each of the four AMD CIDEC variants with mCherry-tagged Perilipin1 (PLIN1) in 3T3-L1 pre-adipocytes. PLIN1 is an adipocyte-specific LD-associated protein that interacts with and potentiates the function of murine CIDEC and hence could be used to track individual LDs [31]. After inducing LD formation with oleic acid treatment, we performed time-lapse imaging over 6 hours to quantify the number of LD fusion events (Fig 4 and S1 Video). Cells expressing each of the AMD CIDEC variants showed significant defects in LD fusion frequency compared to cells expressing CIDEC WT (Student's t-test, p<0.005). Over 6 hours, cells expressing CIDEC WT had 14.6%±1.9% of their LDs achieving fusion (Fig 4A and 4B). Cells expressing CIDEC V47I, Y61H and V161M had a severe decrease in LD fusion events with only 0.3%±0.5%, 2.0%±1.8%

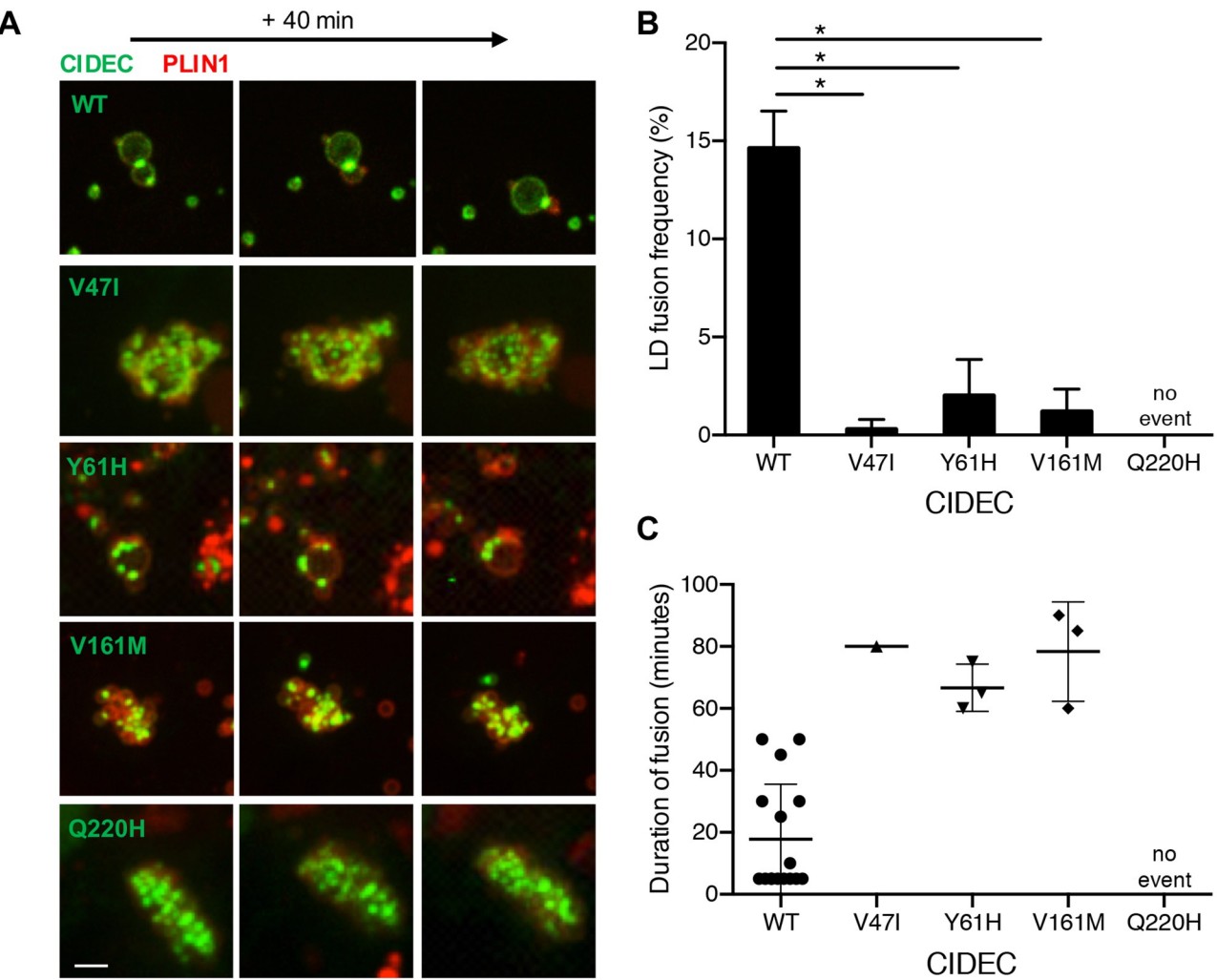

**Fig 4. Lipid droplet (LD) fusion occurs less frequently and more slowly in pre-adipocytes expressing the AMD CIDEC rare variants.** (A) Representative time-lapse images over 40 minutes of LDs in cells co-expressing GFP-tagged CIDEC WT (taken from S1 Video) or each of the rare variants, and mCherry-tagged PLIN1. Scale bar: 2 μm. (B) Percentage of LDs undergoing fusion during the 6-hour analysis. N = 3 (mean ± SD, Student's t test, *p<0.05). (C) Time in minutes required from initial LD–LD contact to complete LD fusion.

and 1.2%±1.1% of LDs achieving fusion respectively. No LD fusion events were recorded during the 6 hours in cells expressing CIDEC Q220H, suggesting that this variant causes a severe loss of LD fusion capacity. Quantification of the time required from initial contact to complete fusion of two LDs revealed that LD fusion events slowdown in presence of the CIDEC variants (Fig 4C). In conclusion, adipocytes expressing the AMD CIDEC variants have a defect in LD fusion capacity, with merging events being slower and rarer than the ones occurring in cells expressing CIDEC WT.

Finally, we performed a Fluorescence Recovery After Photobleaching (FRAP) experiment to determine if the CIDEC variants could affect the kinetics of lipid diffusion between LDs. We transiently transfected 3T3-L1 pre-adipocytes with GFP-tagged CIDEC WT or AMD variants, induced LD formation and labeled LDs with the fluorescent fatty acid BODIPY 558/568 dye. Focusing on adjoining LDs of equivalent size and expressing CIDEC at the contact site, we photobleached one LD and measured the mean optical intensity (MOI) of both the

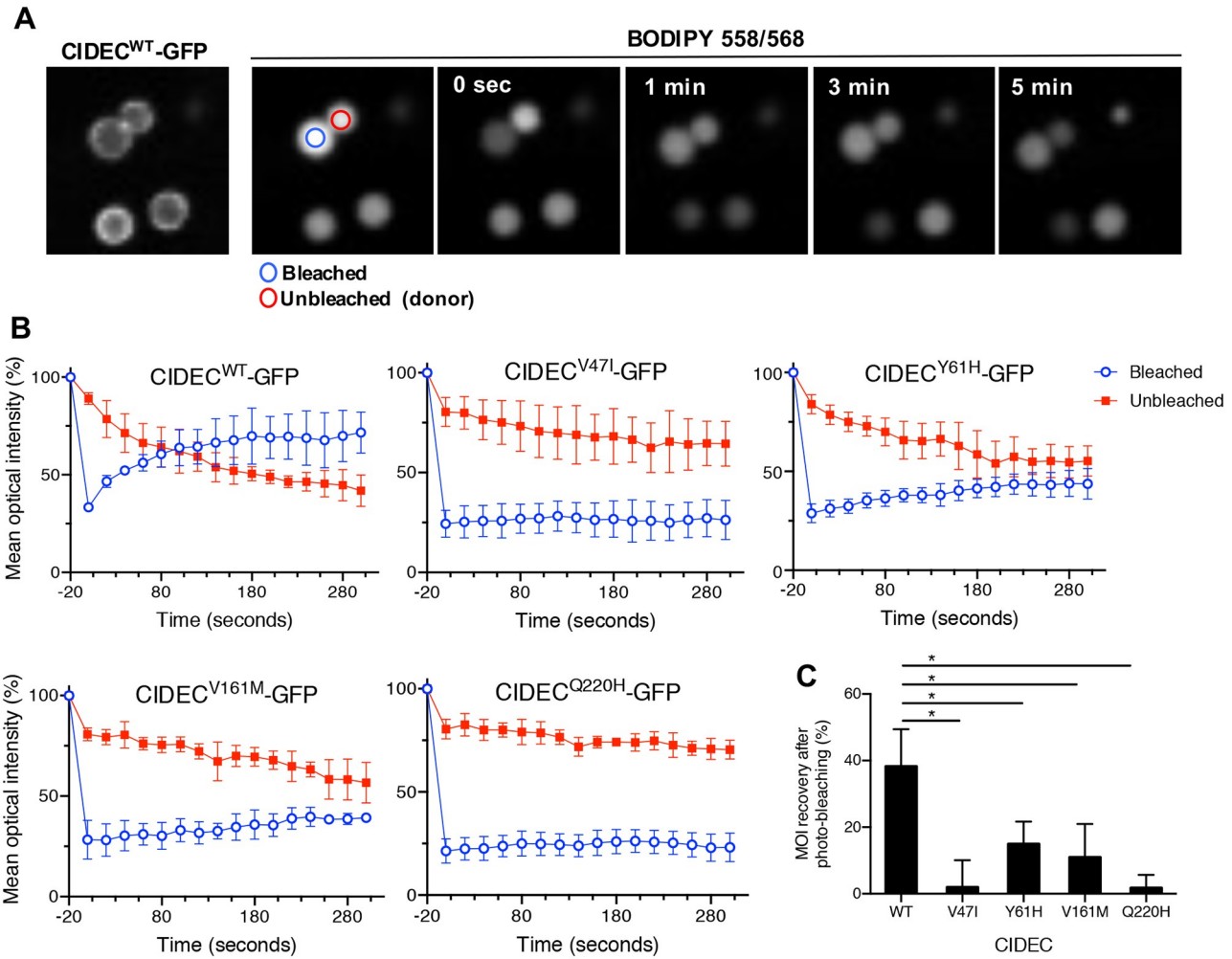

**Fig 5. AMD CIDEC variants cause a decrease in the lipid exchange rate between lipid droplets (LDs).** (A) Representative Fluorescence Recovery After Photobleach (FRAP) images of paired LD expressing GFP-tagged CIDEC wild-type (WT) showing progressive neutral lipid (BODIPY 558/568 dye labeling) exchange as determined by fluorescence recovery from the adjacent LD. (B) Quantification of mean optical intensity (MOI) in the bleached (blue circle) and unbleached (red circle) LD in cells expressing CIDEC WT or each of the rare variants. (C) Percentage of MOI recovery on bleached LDs from 0 sec. to 300 seconds. N = 3 (mean ± SD, Student's t test, *p<0.05).

bleached and the neighboring, unbleached LD. Recovery of fluorescence on the bleached LD over time was used to quantify the rate of lipid exchange between the two LDs (Fig 5A). In cells expressing CIDEC WT, the fluorescence recovered to about 75% of the pre-bleach intensity within 3 minutes in the photobleached LD. This recovery was accompanied by a corresponding decrease in fluorescence on the unbleached LD, reflecting efficient lipid exchange between the two LDs (Fig 5B and 5C). In cells expressing CIDEC Y61H or V161M, fluorescence recovery in the bleached LD exhibited a delayed and reduced fluorescence compared to cells expressing CIDEC WT. In cells expressing CIDEC V47I or Q220H there was very limited, if any, fluorescence recovery on the bleached LD, suggesting loss of lipid exchange capacity (Fig 5B and 5C). In conclusion, the presence of the AMD CIDEC variants impairs the lipid diffusion capacity between LDs in adipocytes.

Collectively, these results show that the AMD CIDEC variants do not affect CIDEC localization to LD contact sites, but they impair the lipid exchange capacity between LDs, resulting

in defective LD fusion and incapacity for the adipocytes to accumulate lipids inside few large LDs.

## Q1 AMD CIDEC rare alleles decrease the binding affinity of CIDEC with the lipid droplet fusion effectors PLIN1, RAB8A and AS160

LDs are highly dynamic organelles containing a neutral lipid core enclosed in a phospholipid monolayer decorated by a large number of proteins [32, 33]. To better understand the functional consequences of the AMD CIDEC alleles and how they can affect LD fusion, we examined if they could alter protein-protein interactions. Indeed, CIDEC-mediated LD fusion is different from other membrane fusion events. CIDEC proteins need to first accumulate at the contact site between two LDs, to then enable recruitment of regulator proteins such as PLIN1, RAB8A and AS160, which facilitate the lipid transfer through the fusion pore [34].

We first assessed if the variants affected CIDEC capacity to homodimerize. CIDEC contains two conserved CIDE domains allowing its dimerization, a N-terminal CIDE-N domain and a C-terminal CIDE-C domain ([23, 35] and Fig 2B). The CIDE-N domain, in which the variants V47I and Y61H are located, dimerizes mainly via electrostatic interactions, while the CIDE-C domain that contains the V161M variant dimerizes through a stronger interaction [31]. Q1 AMD patients are heterozygous for the different CIDEC alleles, so HEK 293T cells were co-transfected with 3xFlag-CIDEC WT and either CIDEC WT, the E186X mutation or one of the AMD CIDEC variants tagged with GFP. After immunoprecipitation of the 3xFlag-CIDEC WT, pulled-down proteins were probed with anti-GFP. Co-transfection of the 3xFlag-CIDEC WT together with GFP alone served as negative control. The CIDE-N domain variants V47I and Y61H showed decreased dimerization capacity with CIDEC WT, whereas the two other variants V161M and Q220H did not affect the binding ability (Fig 6A). The pathogenic mutation CIDEC E186X, located in the CIDE-C domain, also did not affect the binding affinity with CIDEC WT.

Next, we assessed if the AMD CIDEC variants affect CIDEC capacity to interact with its LD-associated regulatory partners PLIN1, RAB8A and AS160, as these interactions are required for LD fusion and growth [31, 36]. We co-transfected HEK 293T cells with 3xFlag-CIDEC WT, E186X or the AMD CIDEC variants, and either PLIN1-mCherry, AS160-GFP or RAB8A-mCherry. After immunoprecipitation of the 3xFlag-CIDEC, pulled-down proteins were probed with anti-mCherry or anti-GFP. Strikingly, all four AMD CIDEC variants had similarly decreased binding affinity with PLIN1 (Fig 6B) and AS160 (Fig 6C). All four AMD CIDEC variants also showed decreased binding capacity with RAB8A, however, the Q220H variant caused a more severe loss of interaction with the GTPase (Fig 6D). Only a fraction of RAB8A and AS160 are associated to LDs, with the rest being distributed in the cytoplasm (Fig 6E and S4 Fig in S1 File). The fact that the E186X mutant is abnormally diffuse in the cytoplasm could explain its stronger interaction with the binding partners compared to CIDEC WT, which is concentrated on the LDs (Fig 6B–6D).

Collectively, these results show that the AMD CIDEC variants V47I and Y61H, located in the CIDE-N domain decrease CIDEC dimerization capacity and its binding ability with the regulators partners PLIN1, RAB8A and AS160. The two other variants, V161M and Q220H, which are not in the CIDE-N domain, do not affect CIDEC ability to dimerize, however, they nevertheless also decrease its interaction with PLIN1, RAB8A and AS160. The reduced interaction capacity of the four AMD CIDEC variants with its binding partners may explain how their presence causes a defect in lipid droplet fusion and enlargement in adipocytes.

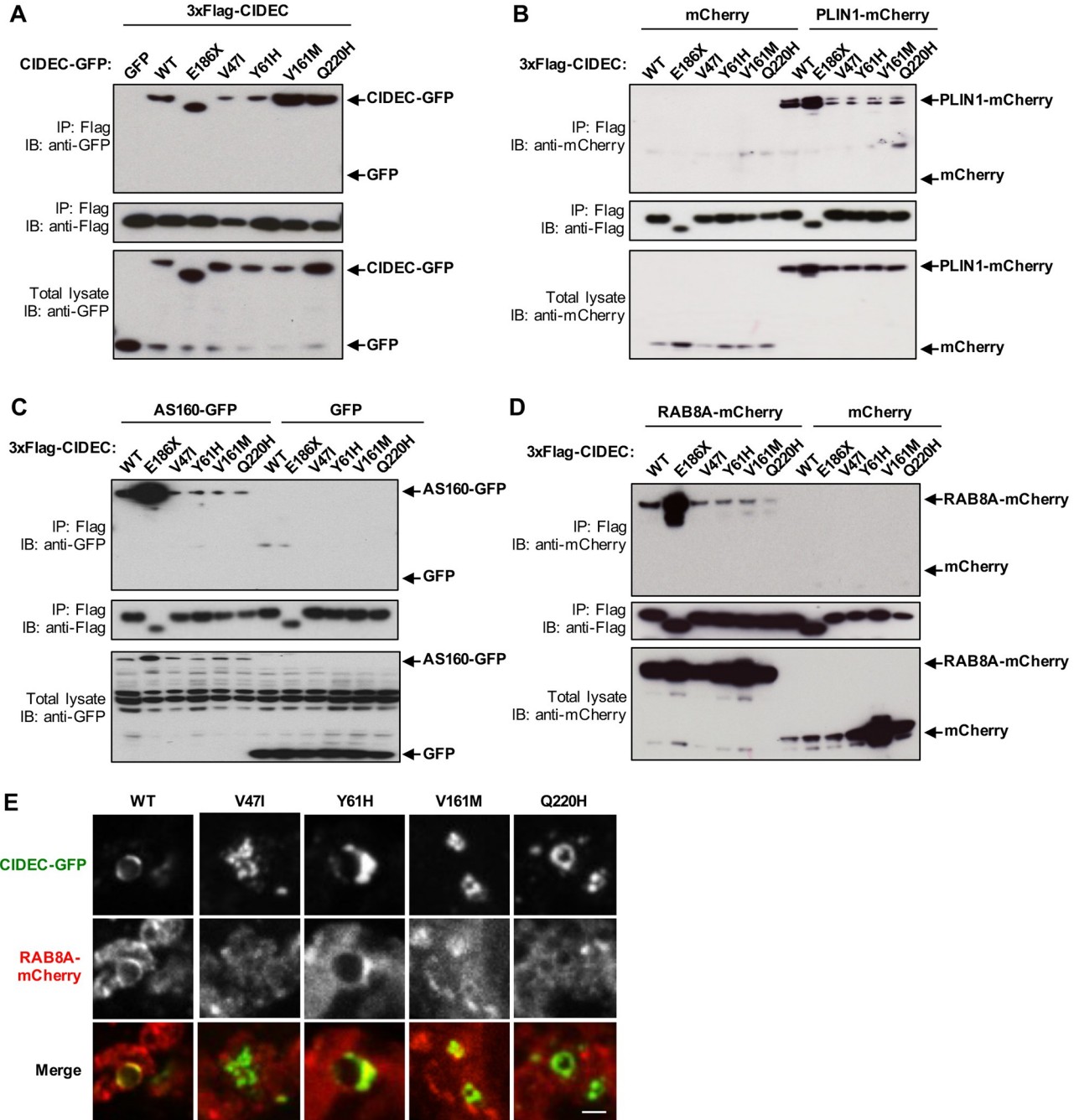

**Fig 6. AMD CIDEC variants in the CIDE-N domain decrease dimerization affinity and all four variants decrease binding to effector partners PLIN1, AS160 and RAB8A.** (A) 3xflag-tagged CIDEC wild-type (WT) was co-transfected with the indicated GFP-tagged CIDEC variants in HEK 293T cells. GFP alone was used as negative control. 3xflag-tagged CIDEC WT was immuno-precipitated (IP) using anti-Flag and pulled-down proteins were immuno-blotted (IB) with anti-GFP and anti-Flag. Total cell lysate was immunoblotted with anti-GFP to control for CIDEC-GFP expression levels. (B-E) HEK 293T cells were co-transfected with 3xFlag-CIDEC WT, E186X or AMD variants, and either PLIN1-mCherry (B), AS160-GFP (C) or RAB8A-mCherry (D). After immunoprecipitation (IP) of the 3xFlag-CIDEC, pulled-down proteins were probed with anti-mCherry or anti-GFP, and anti-Flag. Co-transfection with mCherry or GFP alone was used as negative controls. Total cell lysates were immunoblotted (IB) with anti-mCherry or anti-GFP to control for PLIN1, AS160 and RAB8A expression levels. (E) Representative fluorescence images of 3T3-L1 pre-adipocytes lipid droplets containing CIDEC-GFP wild-type (WT) or variants and RAB8A-mCherry. Scale bar: 2 μm.

## CIDEC expression is not detected in the human retina or the Retinal Pigment Epithelium and the Q1 AMD CIDEC variants do not affect the size of the retinosomes

CIDEC plays a critical role in the white adipose tissue, but is also expressed in organs such as muscles, nerves and even blood vessels [20]. CIDEC expression in the eye has been reported after an Expressed Sequence Tag (EST) database search, however, it is not known if the expression comes from the neuroretina or the eye globe supportive tissue [35]. The key elements of the eye involved in AMD are the photoreceptors, an epithelium located underneath the retina called the Retinal Pigment Epithelium (RPE) and the blood vessels supporting the retina called the choroid. To investigate the potential expression of *CIDEC* in these structures, we first used published human RNA sequencing datasets. *CIDEC* was not detect in human retina or RPE/choroid (bulk RNA sequencing [17]) and in different human ocular cell types (single cell RNA sequencing [18]) (Fig 1C). To investigate further the potential expression of *CIDEC* in the eye, we performed RNA *in situ* hybridization on eye sections from a Caucasian 73-year old female and a Caucasian 88-year old male, both without history of AMD (Fig 7A). We also performed *Fsp27* RNA *in situ* hybridization on mouse eye sections (S5 Fig in S1 File). In both human and mouse eyes, we did not detect CIDEC RNA in the retina, the RPE or the choroid (Fig 7A and S5 Fig in S1 File). We used the sensitive detection method BaseScope™ (Advanced Cell Diagnostics (ACD)) and found that the signal detected with the CIDEC probes on the human eye sections were consistent with the background signal detected using the bacterial gene DapB as a negative control (Fig 7A). On the mouse eye sections, we found rare cells positive for Fsp27 expression but these cells were in the supportive tissue around the eye (S5 Fig in S1 File).

RPE cells contain specific LDs called retinosomes, in which retinyl esters are stored and used to replenish key components of the visual cycle [37, 38]. To account for the possibility that CIDEC was expressed below our detection threshold in RPE cells, we tested if exogenous AMD CIDEC variants could have consequences on the size of these specialized LDs, retinosomes. Primary human fetal RPE cells were infected with lentivirus encoding CIDEC WT or the AMD CIDEC variants and differentiated for three weeks before oleic acid stimulation and LD labelling. Similar to the localization in adipocytes, CIDEC WT and AMD CIDEC variants accumulate on retinosomes and concentrate at the LD fusion sites in RPE cells (Fig 7B). However, the RPE cells failed to form large LDs after oleic acid stimulation and the majority of the retinosomes in RPE cells expressing CIDEC WT were smaller than 1 μm in diameter (Fig 7C). Consequently, we did not observe any difference in LD size between the RPE cells expressing the CIDEC WT and the cells expressing the different AMD CIDEC variants.

Finally, we compared color fundus photos (Fig 8) and Optical Coherence Tomography (OCT) images (not shown) from the eyes of the Q1 AMD CIDEC variant carriers and Q1 AMD CIDEC variant non-carriers. In particular, we wanted to know if by disrupting lipid accumulation, the CIDEC variants could affect size and accumulation of drusen, which are deposits of proteins and lipids building up under the retina and a hallmark of AMD. However, we did not observe unique ocular clinical features in patients carrying the CIDEC rare variants (Fig 8B) compared to non-carriers (Fig 8A). In both groups, we observed typical AMD clinical features such as pigmentary changes, variable amount of drusen, geographic atrophy and choroidal neovascular lesions.

In conclusion, we did not detect CIDEC expression in the ocular structures directly affected in AMD. We also found that exogenous expression of the AMD CIDEC variants did not alter retinosome size in RPE cells, and that AMD patients carrying the CIDEC variants do not present unique phenotypic ocular features compared to non-carriers. Our results suggest that the AMD CIDEC variants do not play a direct role in the eye. Additional experiments using

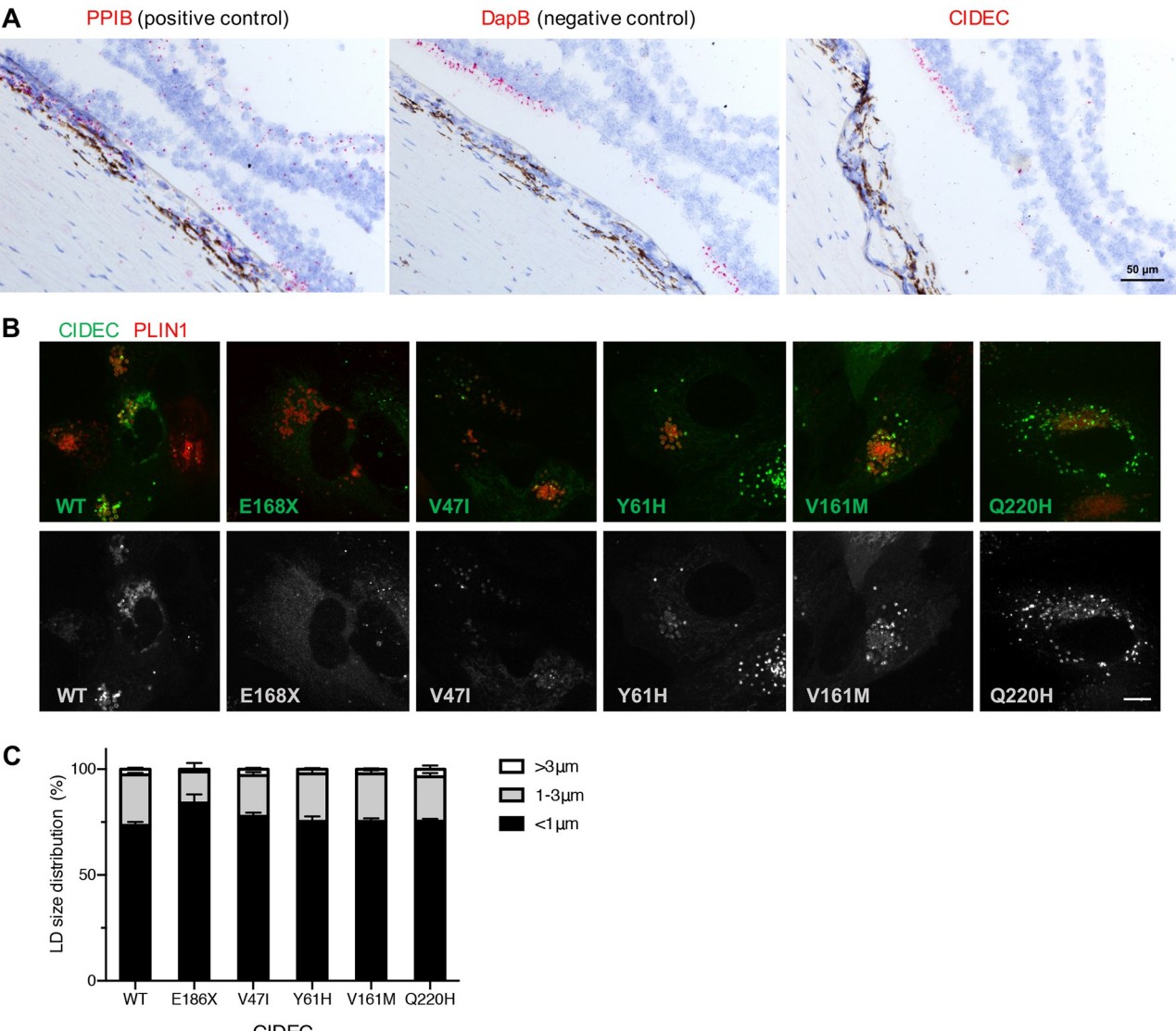

**Fig 7. CIDEC RNA is not detected in the human eye and exogenous expression of the CIDEC variants does not affect lipid droplets (LDs) size in Retinal Pigment Epithelium (RPE) cells.** (A) In situ hybridization in the fovea of a control human donor eye showing that CIDEC RNA is not detected in the retina or RPE cells. Detection of PPIB (red) was used as positive control and detection of bacterial DapB was used as negative control and evaluation of the non-specific background. Scale bar: 50 μm. (B and C) Human fetal RPE cells were co-infected with lentivirus expressing CIDEC variants and PLIN1 as marker for LDs. The infected cells were differentiated for 3 weeks before oleic acid stimulation. Representative images of RPE cells expressing both CIDEC variants and PLIN1 (B). LD diameters were quantified by diameter range as depicted in the bar graph (n = 3; mean ± SD) (C). Scale bar: 5 μm.

conditional mouse models will be important to assigning the tissue specific effects of CIDEC variants and the role of LD dysregulation in AMD.

## Discussion

Here we report the first analysis examining the genetic effect on baseline LLD, a clinical measurement that has been shown to be predictive of anti-VEGF treatment response and GA lesion growth, in AMD patients. While the study is of modest size, to our knowledge, it is novel in its effort to utilize clinical indices beyond BCVA that have been linked to patient

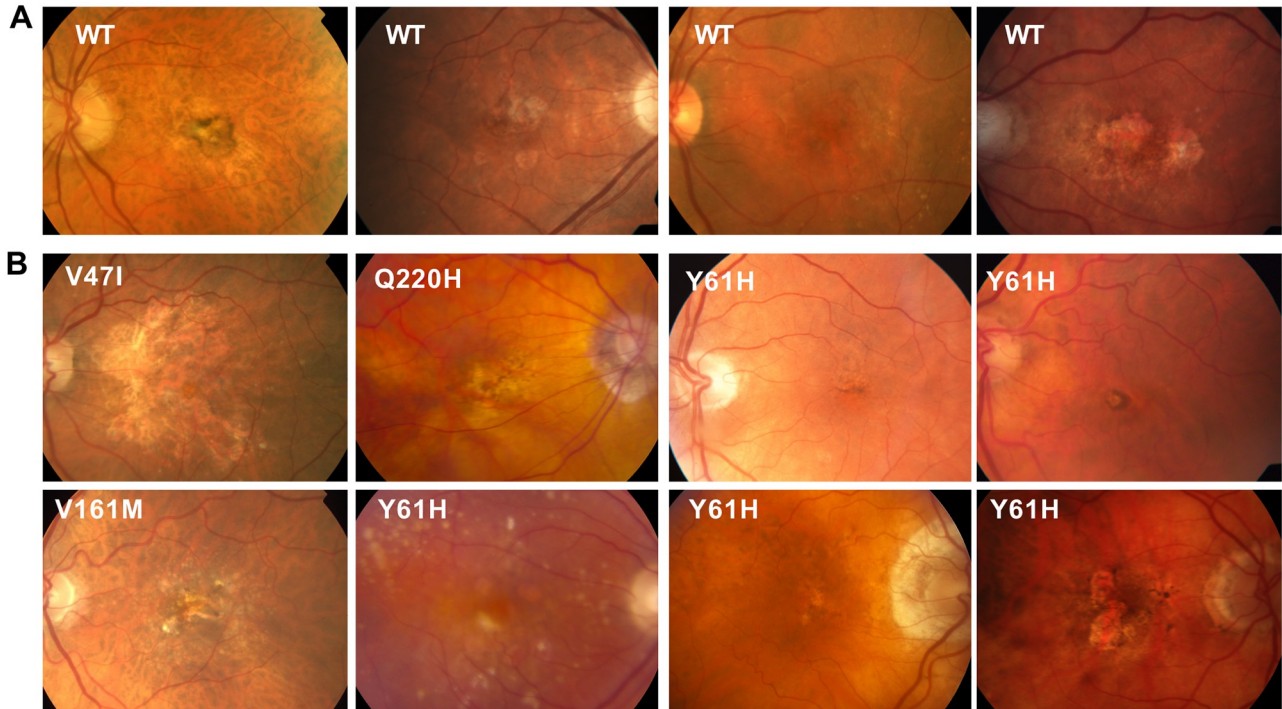

**Fig 8. Clinical images of patients in the low-luminance deficit quartile Q1.** Color fundus photos (CFP) from participants in the HARBOR trial (A) Q1 non-carriers for CIDEC rare variants (B) Q1 rare variant CIDEC carriers. CFP in both groups demonstrate typical clinical features of macular degeneration such as pigmentary changes, drusen, geographic atrophy and choroidal neovascular lesions. No obvious phenotypic differences are noted between the two groups.

outcomes to further homogenize AMD patients in order to increase power for genetic analysis. It is our hope that as datasets increase in size and have deeper phenotypic assessment, these types of sub-phenotype GWAS analyses will increase and work alongside recent studies utilizing novel *in vitro* methods, such as those described here and genome-wide single cell and perturbation methods to help uncover the functionality of genes associated with the pathogenesis of AMD.

We did not find any variants, either in the common or the rare variant burden analysis which passed a pre-specified significance threshold accounting for multiple testing. This can occur for several reasons. One reason could be that the effect of patient germline genetics is not substantial on low-luminance visual acuity and that environmental factors explain more of the risk variability. Another is that even with the three clinical trials used in this study, we are underpowered for a genome wide analysis in this study population. As such, it will be important to replicate the genetic findings in subsequent datasets as they become available with similar phenotyping and sequencing data. Since all patients in this analysis are neovascular AMD patients, and *CIDEC* has not been reported previously as an AMD risk gene, this could indicate that *CIDEC* rare variants play a role only once a patient develops advanced disease. Datasets with deep phenotyping of neovascular AMD patients would be required for replication. While large scale biobank data exist (e.g. U.K. BioBank), and are exceptionally useful for most replication analyses, these datasets do not currently have the ability to delve deep into clinical features for an age-related disease such as AMD. Conversely, smaller, more phenotypically focused genetic datasets such as the ones used in this study are useful for identification of signals and hypotheses, but are severely underpowered to confirm an association statistically. As

such, we sought to assess the possible contribution of *CIDEC*, a gene with biology tangential to genes in known AMD risk loci, to AMD pathology through *in vitro* analysis.

In the rare variant burden analysis, we noticed that one of the top hits was strongly linked with lipid metabolism. Similarly, multiple loci identified in AMD risk studies contain genes implicating lipid metabolism. Due to sample size constraints, and the low frequency of *CIDEC* variants, we were unable to test for interaction between known AMD risk alleles and the *CIDEC* variants. Rare, protein altering variants were enriched in the patients with small LLD. Previous studies associated a small LLD at baseline with more favorable prognostic and predictive outcomes. Thus, possible impairment of CIDEC function could be beneficial for AMD patients and we undertook characterization of the *CIDEC* variants in further studies, which focused on the exact variants seen in our AMD cases.

In our *in vitro* analysis, we interestingly found that all four rare variants failed to impair CIDEC localization to LDs, but instead all decreased the binding affinity of CIDEC with the LD fusion effectors PLIN1, RAB8A and AS160. Interaction of CIDEC with these binding partners is critical for its function and we hypothesize that this decreased interaction underlies the defect in LD enlargement and lipid exchange that we observed in adipocytes expressing the AMD CIDEC variants. Interestingly, the functional consequences that we uncovered are milder than the ones caused by the lipodystrophic E168X variant and are restricted to the LD capacity to fuse and increase lipid storage. Indeed, the AMD CIDEC variants are hypomorphic regarding LD size and they do not affect mitochondria density or activity. Our data suggest that the Q1 AMD CIDEC variants do not severely disrupt adipocyte health and function and may have a beneficial effect by only limiting the capacity of adipocytes to accumulate lipids in very large LDs.

Of note, patients carrying the Q1 AMD CIDEC variants are heterozygotes, contrasting with the E168X homozygote lipodystropic patient. Furthermore, heterozygous Fsp27 wt/ko mice have normal weight and the appearance of their adipose tissue is similar to the one from Fsp27 wt/wt mice [27]. Thus, it is likely that the Q1 AMD CIDEC patients did not suffer from severe lipodystrophy and that they only had sub-clinical consequences of the CIDEC variants expression. Our results suggest that CIDEC is not expressed in the ocular tissue affected in AMD such as the retina, RPE and choroid. This points out toward a "systemic" effect of the beneficial Q1 AMD CIDEC variants. Interestingly, a similar indirect and systemic favorable effect has recently been reported in mouse models of vascular inflammation and atherogenesis after *Fsp27* silencing [39, 40]. Many studies on dietary or circulating lipids, as well as genetic studies support a role for not only local lipid trafficking in the retina but also for circulating lipoproteins in AMD pathogenesis [41]. Therefore, it would have been interesting to perform a biomarker investigation of the HARBOR patient serum to know if any particular change(s) in circulating lipoproteins levels could be detected in patients carrying the Q1 AMD CIDEC variants compared to non-carriers (unfortunately, such samples were not available for us to perform the analysis). In addition to lipoproteins, it would also have been interesting to probe the AMD patient serums for changes in adipokines, the cytokines secreted by adipocytes. Indeed, it has been shown that hypertrophied adipocytes can lead to local inflammation and inhibition of production of adipokines, such as adiponectin [42]. Since adipocytes expressing the AMD CIDEC variants have a decreased LD enlargement capacity, it may prevent them from becoming hypertrophic, keeping adiponectin level high. Supporting this idea, it has been reported that the *Fsp27* deficient mice show increased serum adiponectin level compared to wild type mice [30, 43]. Importantly, increased serum adiponectin levels have been shown to be protective in several pre-clinical models of angiogenesis in the eye, including models of neovascular AMD [44–46] and there is human genetic data linking adiponectin ADIPOQ and its receptor ADIPOR1 to the risk of advanced AMD [47, 48]. Finally, CIDEC, ADIPOQ and APOE (an

AMD GWAS locus also involved in lipid metabolism) have been linked as part of an 8-gene hub identified as candidate serum biomarkers for diabetic peripheral neuropathy [19].

Interestingly, an emerging role for CIDEC as modulator of vascular function and VEGF signaling has been recently proposed [49, 50]. Although in our study, CIDEC mRNA was below detection limit in human retina, RPE and choroid, CIDEC expression in vascular endothelial cells has been observed in other human tissue such as subcutaneous or visceral fat [49]. Using a mouse expressing human CIDEC in vascular endothelial cells, it was shown that CIDEC enhance vascular remodeling, angiogenesis and stabilize VEGF signaling [50]. Since the CIDEC rare variants were identified in neovascular AMD patients, it is unlikely that they critically affect this potential pro-angiogenic role of CIDEC. However, since the patients had better outcome after anti-VEGF therapy, the variants may decrease CIDEC effect on VEGF signaling stabilization, thus potentiating response to treatment.

In conclusion, our rare variant burden genetic analysis followed by our *in vitro* dissection of the functional consequences of the beneficial variants, altogether with published data, suggest that once patients have developed neovascular AMD, the disease outcome could be modified by systemic and indirect lipidomic biological processes that it would be interesting to investigate further. In particular, investigating adipokines serum level, including adiponectin, in AMD patients could provide new biomarkers of neovascular AMD progression or response to anti-VEGF therapy.

## Materials and methods

### Research subjects and low-luminance deficit (LLD)

The HARBOR clinical trial (ClinicalTrials.gov identifier: NCT00891735) was a 24-month Phase III study designed to evaluate the effectiveness of monthly or as-needed ranibizumab delivery in patients with subfoveal neovascular AMD. This study has been described previously and, after approval by applicable institutional review boards, was conducted in accordance with Good Clinical Practice, applicable FDA regulation and the Health Insurance Portability and Accountability Act [13, 15].

TENAYA (ClinicalTrials.gov identifier: NCT03823287) and LUCERNE (ClinicalTrials.gov identifier: NCT03823300) were randomized, double-masked, non-inferiority trials designed to investigate the efficacy, durability, and safety of intravitreal faricimab in patients with neovascular AMD. Faricimab is a bispecific antibody acting through dual inhibition of angiopoietin-2 and vascular endothelial growth factor A. The trials enrolled 1329 patients and followed them for 112 weeks. These studies have been described previously and, after approval by appropriate regulatory authorities, applicable institutional review boards, and ethics committees, were conducted in accordance with the Declaration of Helsinki and principles of Good Clinical Practice [14].

LLD dysfunction is quantified by first assessing best corrected visual acuity (BCVA) under normal lighting conditions, followed immediately by a low-luminance visual acuity (LLVA) measurement, and this has been described previously [11, 12].

All patient data was anonymized prior to genetic analysis. Before execution of this study, an internal Genentech team of ICF experts reviewed the ICFs from all the clinical studies to ensure appropriate use of the samples.

### Whole genome sequence data quality control and analysis

Subjects who agreed to an informed written consent form for genetic analysis from the above clinical studies were included in the genetic analysis and their DNA samples were sent for WGS. The WGS data for the discovery study population was generated to a read depth of 30X

using the HiSeq platform (Illumina X10, San Diego, CA, USA) processed using the Burrows-Wheeler Aligner (BWA) / Genome Analysis Toolkit (GATK) best practices pipeline. WGS short reads were mapped to hg38 / GRCh38 (GCA_000001405.15), including alternate assemblies, using BWA version 0.7.9a-r786 to generate BAM files. All WGS data was subject to quality control and checked for concordance with SNP fingerprint data collected before sequencing. After filtering for genotypes with a GATK genotype quality greater than 90, samples with heterozygote concordance with SNP chip data of less than 75% were removed. Sample contamination was determined with VerifyBamID software, and samples with a freemix parameter of more than 0.03 were excluded. Joint variant calling was done using the GATK best practices joint genotyping pipeline to generate a single variant call format (VCF) file. The called variants were then processed using ASDPEx to filter out spurious variant calls in the alternate regions.

Sample genotypes were set to missing if the Genotype Quality score was less than 20 and SNPs were removed if the missingness was higher than 5%. SNPs were filtered if the significance level for the Hardy-Weinberg equilibrium test was less than $5x10^{-8}$. The allele depth balance test was performed to test for equal allele depth at heterozygote carriers using a binomial test (discovery WGS only); SNPs were excluded if the p-value was less than $1x10^{-5}$.

Logistic regression was used to assess the association in the common variant analysis, adjusted for age, sex, baseline visual acuity and genetically determined ancestry. PLINK version 1.90b3.46 was used for the common variant analysis.

The sequence data was annotated using SnpEff and there were 120,580 exonic coding variants at a minor allele frequency < 1%. For a gene to be included in the analysis, it had to contain at least two coding SNPs, resulting in 13,046 genes that could be tested for rare-variant gene-burden and a Bonferroni corrected P-value of $3.83x10^{-6}$. The rare variant gene burden test was used to assess the cumulative effect of rare variants. Rvtest software (version 20170228) was used for the combined multivariate and collapsing gene burden test, adjusted for age, sex, baseline visual acuity and genetically determined ancestry. The rare variant gene burden test was used to assess the cumulative effect of rare variants (MAF < 1%).

## Patient population selection

As we are looking at baseline characteristics, all HARBOR patients, regardless of randomized treatment assignment, were eligible for inclusion in our study. We excluded patients who did not consent for exploratory analyses and patients of non-European descent. This resulted in the removal of 118 individuals from the overall enrolled trial study population, representing 10.7% of the study population. We stratified patients for analysis based on LLD quartile 1 (Q1) vs quartile 4 (Q4) as described previously [12]. This resulted in 275 patients in our Q1 group, and 241 patients in Q4. Further demographic information for HARBOR Q1 and Q4 patients is found in Table 1. The same patient population selection criteria were applied to the TENAYA and LUCERNE patients resulting in 171 patients in the Q1 group and 170 patients in the Q4 group. Further demographic information for TENAYA and LUCERNE Q1 and Q4 patients is found in S2 Table in S1 File.

## Cell culture and treatments

293T cells and 3T3-L1 preadipocytes were cultured in Dulbecco's modified Eagle's Medium (DMEM) containing 10% fetal bovine serum (FBS) and Penicillin (10,000 units/ml)/ Streptomycin (10,000μg/ml, 1:100 dilution of stock, Gibco #15140–122). After reaching confluency, 3T3-L1 pre-adipocytes were cultured for 48 hours in DMEM + 10% FBS. The culture medium was then replaced by DMEM + 10% FBS + 5 μg/ml insulin (Sigma, I0516) + 1 μM

dexamethasone (G-bioscience, API-04) + 0.5 mM isobutylmethylxanthine (Sigma, I5879)] to induce adipocyte differentiation. After 48 hours, the medium was replaced by DMEM + 10% FBS + 5 µg/ml insulin for an additional 48–72 hours to achieve complete differentiation. To induce lipid droplet formation, cells were treated with 200 µM of oleic acid-albumin from bovine serum (Sigma, O3008). Human fetal retinal pigment epithelial cells (hfRPE, Lonza, #00194987) were cultured in RtEGM with supplement medium as indicated by the manufacturer's protocol (RtEGM bullet kit, Lonza, #00195409). HfRPE cells were cultured to high confluence on coverglass culture plates (Thermo, 155411) for three weeks to obtain polarized RPE monolayers. After differentiation, hfRPE cells were treated with 20µM A2E (N-Retinylidene-N-Retinylethanolamine, 20mM stock dissolved in DMSO, Gene and Cell Technologies) for 24 hours.

## Plasmids, transfection and viruses

3x Flag- and GFP-tagged expression plasmids were used to express human CIDEC wild type (WT) and the CIDEC rare variants (E186X, V47I, Y61H, V161M, or Q220H) (Genecopoeia, Inc.). GFP- and mCherry-tagged plasmids were used to express human PLIN1, AS160, and RAB8A (Genecopoeia, Inc.). 293T and 3T3-L1 cells were transiently transfected using Lipofectamine2000 and Lipofectamine3000 (Invitrogen, 11668 and L3000). Expression of GFP-tagged CIDEC in hfRPE cells was carried out using lentivirus infection. Viral media were collected from 293T cells transiently transfected with viral vector (expression plasmid), delta8.9, and VSV-G in a molar ratio of 1:2.3:0.2 using Lipofectamine2000. HfRPE cells were infected (without polybrene) on the day when they were split onto 6 well culture apparatuses and kept in viral media for 4–5 days.

## Immunofluorescence, immunoprecipitation and immunoblotting

Anti-Flag (Sigma, F7425), anti-GFP (Abcam, ab6556), anti-mCherry (Abcam, ab167453) antibodies were obtained from commercial sources. Alexa 488-, Alexa-594-conjugated secondary antibodies were obtained from Invitrogen. HRP-labeled secondary antibodies were purchased from Cell Signaling Technologies. Cells were fixed with 4% Paraformaldehyde (EMS, 15710S) for 15 minutes and mounted using ProLong Gold anti-fade mounting medium with DAPI (Thermo Scientific, P36941). Images were obtained with a Nikon A1R confocal microscope or Yokogawa CSU-X spinning disk on a Nikon TiE microscope and a Photometrics Prime 95B. Image acquisition was performed using the NIS elements software 4.50 (Nikon). Co-immunoprecipitation was performed on 293T cells lysed in IP Lysis buffer (Pierce #87788) containing a proteasome inhibitor cocktail (Pierce, Thermo Scientific, 87788) two days after transient transfection. The cell lysates were incubated with anti-Flag M2 affinity beads (Sigma, F2426) overnight at 4˚C. After pull-down of the agarose beads, the immunoprecipitates were washed three times with IP Lysis buffer and eluted in a 2x BOLT Lithium dodecyl sulfate sample buffer for Western blot analysis. The samples were electrophoresed on NuPage 4–12% Bis-Tris gels (Invitrogen #NP0303) in MES-SDS running buffer (Invitrogen, #NP0002) and transferred to PVDF membrane (Invitrogen, #IB24001) for immunoblotting.

## Lipid droplet (LD) assays

3T3-L1 preadipocytes were fixed, stained with the LD marker Bodipy 558/568 C12 fatty acid (Molecular Probes, D3835) and LD diameters were measured in 100 to 150 cells from three independent experiments using Imaris software (Bitplane) and Matlab image processing toolkit. For live cell imaging, 3T3-L1 preadipocytes were transiently co-transfected with GFP-tagged CIDEC and PLIN1-mCherry as LD markers, and incubated with 200 µM of oleic acid.

Images were taken using the Nikon TiE spinning disk confocal microscope with an environmental chamber (Okolab) for 12 hours in 5-minute intervals. The frequency of LD fusion per cell and the time duration of LD fusion from three independent experiments were quantified and plotted using Microsoft Excel 2011 and Graphpad Prism version 8.0.1. Fluorescence Recovery After Photobleaching (FRAP)-based lipid diffusion assays were conducted on the Nikon A1R confocal microscope. FRAP was performed on 3T3-L1 preadipocytes transiently transfected with GFP-tagged CIDEC were incubated with 200 μM of oleic acid and stained with Bodipy 558/568 C12 fatty acid (Molecular Probes, D3835) for 15 hours. One hour before the beginning of the FRAP assay, the medium was changed. LD pairs with clear GFP expression at the contact sites were selected for bleaching. Selected regions were bleached with a 561 mm laser at 100% power for 62.4 milliseconds, followed by time-lapse scanning of 20-second intervals. Mean optical intensity (MOI) of the bleached and the unbleached adjacent LD was measured by ImageJ and plotted using Microsoft Excel 2011 and Graphpad Prism version 8.0.1.

## Mitochondria assays

3T3-L1 cells expressing GFP-tagged human CIDEC were incubated with MitoTracker (#M7512; Thermo Fisher Scientific) before fixation, followed by permeabilization with 0.5% Triton X-100 and staining with DAPI. The mitochondrial density of the CIDEC-expressing cells was determined by measuring the fluorescent intensity of the MitoTracker signal using ImageJ. For the Seahorse Cell Mito Stress Test, 3T3-L1 cells expressing GFP or GFP-tagged CIDEC WT and rare variants (V47I, Y61H, V161M, Q220H, and E186X) were plated on a 96-well assay plate ($10^4$ cells/well). The cells were maintained in XF assay medium (Agilent, #102365100) and subjected to a mitochondrial stress test, using the extracellular flux assay kit by sequentially applying oligomycin (2 mmol/L), carbonyl cyanide 4-(trifluoromethoxy) phenylhydrazone (FCCP; 5 mmol/L), and antimycin/rotenone (1 mmol/L and 1 mmol/L) (Cell Mito Stress Test Kit, Agilent, #103015100). Analysis was carried out by using the Seahorse analyzer software.

## In situ hybridization

The *in situ* hybridization (ISH) BaseScope™ v2 assay (Advanced Cell Diagnostics (ACD)) was performed on 5 μm-thick formalin-fixed paraffin-embedded sections of adult human eyes according to the BaseScope™ detection reagent kit v2 ACD protocol. Probes against the ubiquitously expressed isomerase PPIB were used as positive control, and probes against bacterial DapB were used as negative control. Six custom probes of 18–25 bp oligonucleotide sequences were designed by ACD for highly specific and sensitive detection of human CIDEC RNA. After deparaffinization in xylene and endogenous peroxidase activity inhibition by $H_2O_2$ (10 min), sections were permeabilized and submitted to heat (15 min at 100°C) and protease IV treatment (20 min at 40°C). After probe hybridization for 2 hours at 40°C, the signal was chemically amplified using the kit reagents and detected using the FastRED dye. The sections were then counterstained with Hematoxylin and mounted using VectaMount (Vector Labs, H-5000).

## Clinical images

As part of the HARBOR clinical trial (NCT00891735) [13], color fundus photographs, fluorescein angiography, and spectral-domain optical coherence tomography images (Cirrus; Carl Zeiss Meditec, Inc., Dublin, CA) were collected.

## Statistics for the *in vitro* analysis

Quantification of data was reported as the means ± standard deviation for the indicated number of experiments. Statistical significance of continuous data was tested by the two-tailed Student's t-test. $p < 0.05$ was considered statistically significant.

## Web resources

dbSNP: https://www.ncbi.nlm.nih.gov/snp
  Ensembl: http://grch37.ensembl.org/Homo_sapiens/Info/Index
  OMIM: http://www.omim.org/
  Uniprot: https://www.uniprot.org/
  GTEx: https://www.gtexportal.org/home/
  PolyPhen2: http://genetics.bwh.harvard.edu/pph2/
  Genebass: https://genebass.org/
  rvtest: http://zhanxw.github.io/rvtests/
  Snpeff: http://pcingola.github.io/SnpEff/
  gnomAD: https://gnomad.broadinstitute.org/

## Supporting information

**S1 File.**
(PDF)

**S1 Raw images. Uncropped scans of the films used to build Fig 6A–6D.** The area used in Fig 6 are highlighted on each film by red rectangles.
(PDF)

**S1 Video. Example of a representative time-lapse video of a lipid droplet fusion event in pre-adipocytes expressing GFP-CIDEC wild-type (green) and mCherry-tagged PLIN1 (red).**
(MP4)

**S1 Fig.**
(PNG)

## Acknowledgments

We thank all of our Genentech colleagues involved in the Human Genetics Initiative including Julie Hunkapiller, Jens Reeder, and Suresh Selvaraj. We also thank our colleagues in the Research Pathology department, including Patrick Caplazi and Susan Haller.

## Author Contributions

**Conceptualization:** Sehyun Kim, Amy Stockwell, Ivaylo Stoilov, Brian L. Yaspan, Marion Jeanne.

**Data curation:** Amy Stockwell.

**Formal analysis:** Sehyun Kim, Amy Stockwell, Arthur Wuster, Brian L. Yaspan, Marion Jeanne.

**Investigation:** Sehyun Kim, Amy Stockwell, Han Qin, Simon S. Gao, Arthur Wuster, Phillip Lai, Brian L. Yaspan, Marion Jeanne.

**Methodology:** Sehyun Kim, Amy Stockwell, Han Qin, Meredith Sagolla, Arthur Wuster, Phillip Lai, Brian L. Yaspan, Marion Jeanne.

**Project administration:** Marion Jeanne.

**Resources:** Simon S. Gao, Meredith Sagolla, Ivaylo Stoilov.

**Supervision:** Brian L. Yaspan, Marion Jeanne.

**Validation:** Brian L. Yaspan, Marion Jeanne.

**Visualization:** Sehyun Kim, Amy Stockwell, Arthur Wuster.

**Writing – original draft:** Brian L. Yaspan, Marion Jeanne.

**Writing – review & editing:** Sehyun Kim, Ivaylo Stoilov.

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
