## [Decision Letter · Decision Letter 0]

27 Dec 2022

PONE-D-22-30538Rare CIDEC coding variants enriched in Age-related Macular Degeneration patients with small low-luminance deficit cause lipid droplet and fat storage defectsPLOS ONE

Dear Dr. Marion,

Thank you for submitting your manuscript to PLOS ONE. After careful consideration, we feel that it has merit but does not fully meet PLOS ONE’s publication criteria as it currently stands. Therefore, we invite you to submit a revised version of the manuscript that addresses the points raised during the review process.

In particular what was underlined by Reviewer 1.

We look forward to receiving your revised manuscript.

Kind regards,

Giuseppe Novelli

Academic Editor

PLOS ONE

Journal Requirements:

3 . Thank you for stating the following in the Competing Interests section: 

"I have read the journal's policy and the authors of this manuscript have the following competing interests: at the time of the study, all authors were full time employees of Genentech/Roche with stock and stock options in Roche."

4.In your Data Availability statement, you have not specified where the minimal data set underlying the results described in your manuscript can be found. PLOS defines a study's minimal data set as the underlying data used to reach the conclusions drawn in the manuscript and any additional data required to replicate the reported study findings in their entirety. All PLOS journals require that the minimal data set be made fully available. For more information about our data policy, please see http://journals.plos.org/plosone/s/data-availability.

Reviewers' comments:

Reviewer's Responses to Questions

**Comments to the Author**

1. Is the manuscript technically sound, and do the data support the conclusions?

Reviewer #1: Yes

Reviewer #2: Yes

2. Has the statistical analysis been performed appropriately and rigorously? 

Reviewer #1: Yes

Reviewer #2: Yes

3. Have the authors made all data underlying the findings in their manuscript fully available?

Reviewer #1: Yes

Reviewer #2: Yes

4. Is the manuscript presented in an intelligible fashion and written in standard English?

Reviewer #1: Yes

Reviewer #2: Yes

5. Review Comments to the Author

Reviewer #1: The authors are invited to extend their analysis and to resubmit the article with a much larger sample that more robustly supports the association between the genetic variants and the clinical findings.

Reviewer #2: The authors Kim et al. performed genome wide analysis to identify genetic determinants of low-luminance vision deficit in patients with age-related macular degeneration (AMD). They identified four coding variants in the CIDEC gene and carried out in vitro their functional characterization related to lipid metabolism. The results suggest the absence of their direct role in the eye.

Major comments

The authors definy the variants identified in the CIDEC gene as rare variants, without reporting their eventually known frequency in the population genetic studies (i.e. the gnomAD database could serve as useful reference of allelic frequency).

The authors classified three of the four CIDEC variants as probably/possibly damaging using a single tool Polyphen to predict in silico possible impact of the aminoacid substitutions on the structure and function of the CIDEC protein. American College of Medical Genetics and Genomics (ACMG) developed the guidelines for the interpretation of sequence variants and Varsome represent a bioinformatic tool for variant classifications according to ACMG criteria. For example, your identified V47I varianti is classified as benign using Varsome, as likely benign in ClinVar database and the allele frequency in gnomAD is 0.001721.

The criteria of variants classifications are important in their functional characterization because the authors don’t discuss why the results demonstrated the same decrease of dimerization capacity of V47I substitution predicted to be probably damaging and Y61H substitution predicted to be benign or the same reduced lipid exchange between two lipid droplet after transfection of the benign Y61H substitution or the possibly damaging V161M substitution.

Finally, the authors conclude that the CIDEC variants do not play a direct role in the eye and influence low-luminance vision deficit via an indirect and systemic effect related to fat storage capacity.

A recent paper (Balakrishnan et al., Diabetes. 2022 Oct 18:db220294. doi: 10.2337/db22-0294) described the CIDEC expression in human endothelial cells (ECs) regulating vascular function. Could be a new working hypothesis to explain the genetic molecular data in patients with AMD?

6. PLOS authors have the option to publish the peer review history of their article (what does this mean?). If published, this will include your full peer review and any attached files.

Reviewer #1: No

Reviewer #2: No

---

## [Author Response · Author response to Decision Letter 0]

28 Mar 2023

I. Uploaded during the resubmission:

- A rebuttal letter labeled 'Response to Reviewers'.

- A marked-up copy of the manuscript labeled 'Revised Manuscript with Track Changes', with changes tracked and highlighted in color.

- An unmarked version of the revised paper without tracked changes labeled 'Manuscript'.

II. Amended Statements to adhere to the Journal Requirements:

- Financial disclosure: 

“No external funding was obtained for this study. At the time of the study, all authors were full time employees of Genentech/Roche. The study was funded by general Genentech/Roche funding. The funders had no role in study design, data collection and analysis, decision to publish, or preparation of the manuscript.”

- Competing Interests section: 

“I have read the journal's policy and the authors of this manuscript have the following competing interests: at the time of the study, all authors were full time employees of Genentech/Roche with stock and stock options in Roche. This does not alter our adherence to PLOS ONE policies on sharing data and materials.”

- Data Availability statement:

“Study consents and contractual obligations regarding the generation of whole genome sequencing data prevent posting data for download. A request for the summary statistics from the genetics analysis used in this study can be directed to hg-data-request-d@gene.com”

- Original uncropped and unadjusted images:

“Original uncropped and unadjusted images underlying all blot results are available and provided in Supporting Information.”

- “Data not shown” in your manuscript:

A citation to support this sentence was already present in the original manuscript: Line 249: “As expected, the mutant CIDEC E168X was diffused in the cytoplasm and failed to accumulate around the LDs (data not shown, and [29]).” We can only keep the citation and remove the “data not shown” mentioned in that sentence if it is preferred.

III. Response to Reviewers' comments:

1. Is the manuscript technically sound, and do the data support the conclusions?

Reviewer #1: Yes

Reviewer #2: Yes

2. Has the statistical analysis been performed appropriately and rigorously?

Reviewer #1: Yes

Reviewer #2: Yes

3. Have the authors made all data underlying the findings in their manuscript fully available?

Reviewer #1: Yes

Reviewer #2: Yes

4. Is the manuscript presented in an intelligible fashion and written in standard English?

Reviewer #1: Yes

Reviewer #2: Yes

5. Review Comments to the Author

Reviewer #1: The authors are invited to extend their analysis and to resubmit the article with a much larger sample that more robustly supports the association between the genetic variants and the clinical findings.

We agree with the reviewer that it would be optimal to have a larger dataset that more robustly supports the genetic association between the variants found in CIDEC and the clinical findings. However, we used data from 3 clinical trials that are, to our knowledge, the only studies worldwide that have the available sub-phenotyping clinical data in concert with the whole genome sequencing (or exome sequencing) that is required to assess rare variation genome wide. We understood this at the outset of the study, which is why a detailed in vitro assessment of the variants was undertaken.

Reviewer #2: The authors Kim et al. performed genome wide analysis to identify genetic determinants of low-luminance vision deficit in patients with age-related macular degeneration (AMD). They identified four coding variants in the CIDEC gene and carried out in vitro their functional characterization related to lipid metabolism. The results suggest the absence of their direct role in the eye.

Major comments

- The authors definy the variants identified in the CIDEC gene as rare variants, without reporting their eventually known frequency in the population genetic studies (i.e. the gnomAD database could serve as useful reference of allelic frequency).

We thank the reviewer for pointing out this omission. We calculated the rarity of the variants in our dataset consisting of European ancestry AMD patients, but agree it is necessary to point out the frequency in world super populations. We have amended the text (lines 184-189) to read:

“Three of these SNPs (V47I, V161M and Q220H) are rare (MAF <0.01) in all super-populations seen in gnomAD. Y61H is rare (MAF <0.01) in the European and East Asian super populations, but common (MAF>0.01) in others (South Asian, Ashkenazi Jewish, African/African American and Latinx/Admixed American), with the frequency in the South Asian population being the greatest (MAF=0.07).”

- The authors classified three of the four CIDEC variants as probably/possibly damaging using a single tool Polyphen to predict in silico possible impact of the aminoacid substitutions on the structure and function of the CIDEC protein. American College of Medical Genetics and Genomics (ACMG) developed the guidelines for the interpretation of sequence variants and Varsome represent a bioinformatic tool for variant classifications according to ACMG criteria. For example, your identified V47I varianti is classified as benign using Varsome, as likely benign in ClinVar database and the allele frequency in gnomAD is 0.001721.

The criteria of variants classifications are important in their functional characterization because the authors don’t discuss why the results demonstrated the same decrease of dimerization capacity of V47I substitution predicted to be probably damaging and Y61H substitution predicted to be benign or the same reduced lipid exchange between two lipid droplet after transfection of the benign Y61H substitution or the possibly damaging V161M substitution.

We only used Polyphen as in silico tool to interrogate the potential effect of the four variants as no structure of the full CIDEC protein is available (Polyphen then relies on conservation of the residues across species). Also, the variants were identified in patients with a better outcome of the disease, therefore, prediction based on general population frequency may not be as useful as for variants for which a deleterious effect is expected. We ultimately decided to move on to the functional in vitro analysis irrespective of the in silico predictions. Interestingly, we found that all variants cause a similar hypomorphic phenotype on fat storage in lipid droplets and only affect some of CIDEC functions. For example, all variants have no effect on mitochondria activity, all variants impair lipid exchange capacity, but only 2 out of 4 variants affect CIDEC dimerization. We found that variants can impair fat storage in lipid droplets in 3 different ways: 1. By affecting localization to lipid droplets (E168X, present in the lipodystrophy patient), 2. By affecting CIDEC dimerization (V47I and Y61H) 3. By affecting binding to effector proteins PLIN1/RAB8A/AS160 (V161M and Q220H). The fact that the variants have different effects (benign or damaging) depending on the cellular function assessed may explain why different predictive tools give contradicting conclusions. 

- Finally, the authors conclude that the CIDEC variants do not play a direct role in the eye and influence low-luminance vision deficit via an indirect and systemic effect related to fat storage capacity.

A recent paper (Balakrishnan et al., Diabetes. 2022 Oct 18:db220294. doi: 10.2337/db22-0294) described the CIDEC expression in human endothelial cells (ECs) regulating vascular function. Could be a new working hypothesis to explain the genetic molecular data in patients with AMD?

Absolutely. Thank you for bringing to our attention this recent publication that was not yet available when we wrote the manuscript. When examining the neuroretina, the retinal pigment epithelium and the choroid (the blood vessels supporting the retina and involved in AMD), CIDEC expression at the mRNA level was below detection levels by single cell RNA sequencing, bulk RNA sequencing and in situ hybridization. However, we cannot totally exclude that CIDEC protein is produced by vascular endothelial cells of AMD patients and influences vascular remodeling and angiogenesis. 

In our study, the 4 CIDEC variants were identified in neovascular AMD patients so it is unlikely that they strongly affect CIDEC role in promoting angiogenesis and VEGF signaling as pathologic neovascularization was present. However, patients carrying the variants had better outcomes after anti-VEGF therapy so the variants may decrease CIDEC ability to stabilize VEGF and activate VEGF signaling, thus potentiating treatment response. We added a paragraph in the discussion to mention this new potential direct link between CIDEC variants and AMD outcome via VEGF signaling. See line 539-548.

Of note, Balakrishnan et al. found that endothelial CIDEC, in addition to playing a role on vascular function, also influences metabolic dysfunction, supporting our hypothesis of CIDEC influencing AMD outcome via a systemic effect related to fat storage. In conclusion, CIDEC is now linked to metabolic and vascular biology, which have both been implicated in AMD pathophysiology, strengthening the relevance of our study and findings.

6. PLOS authors have the option to publish the peer review history of their article (what does this mean?). If published, this will include your full peer review and any attached files.

Do you want your identity to be public for this peer review? For information about this choice, including consent withdrawal, please see our Privacy Policy.

Reviewer #1: No

Reviewer #2: No

---

## [Editor Report · Decision Letter 1]

6 Apr 2023

Rare CIDEC coding variants enriched in Age-related Macular Degeneration patients with small low-luminance deficit cause lipid droplet and fat storage defects

PONE-D-22-30538R1

Dear Dr. Jeanne,

We’re pleased to inform you that your manuscript has been judged scientifically suitable for publication and will be formally accepted for publication once it meets all outstanding technical requirements.

Kind regards,

Giuseppe Novelli

Academic Editor

PLOS ONE
---

## [Editor Report · Acceptance letter]

11 Apr 2023

PONE-D-22-30538R1 

Rare CIDEC coding variants enriched in Age-related Macular Degeneration patients with small low-luminance deficit cause lipid droplet and fat storage defects 

Dear Dr. Jeanne:

I'm pleased to inform you that your manuscript has been deemed suitable for publication in PLOS ONE. Congratulations! Your manuscript is now with our production department. 

Kind regards, 

on behalf of

Prof. Giuseppe Novelli 

Academic Editor

PLOS ONE